# The influence of LncRNA H19 polymorphic variants on susceptibility to cancer: A systematic review and updated meta-analysis of 28 case-control studies

**Kunpeng Wang[1], Zheng Zhu[2], Yiqiu Wang[3], Dayuan Zong[1], Peng Xue[1], Jinbao Gu[1], Daoyuan Lu[1], Chuanquan Tu[1]***

**1** Department of Urology, The First People's Hospital of Lianyungang, Lianyungang Clinical Medical College of Nanjing Medical University, Lianyungang, China, **2** First Clinical Medical College of Nanjing Medical University, Nanjing, China, **3** Department of Surgical Oncology, Xuzhou Central Hospital, Southeast University Cancer Institute, Xuzhou, China

☯ These authors contributed equally to this work.
* tuchuanquan1116@126.com

**Data Availability Statement:** All relevant data are within the manuscript and its Supporting Information files.

## Abstract

### Objective

Although myriad researches upon the associations between LncRNA H19 polymorphic variants (rs2839698 G>A, rs217727 G>A, rs2107425 C>T, rs2735971 A>G and rs3024270 C>G) and the susceptibility to cancer have been conducted, these results remained contradictory and perplexing. Basing on that, a systematic review and updated meta-analysis was performed to anticipate a fairly precise assessment about such associations.

### Methods

We retrieved the electronic databases EMBASE, PubMed and Web of Science for valuable academic studies before February 28, 2021. Ultimately, 28 of which were encompassed after screening in this meta-analysis, and the available data was extracted and integrated. The pooled odds ratios (ORs) with 95% confidence intervals (CIs) was used to evaluate such associations. For multi-level investigation, subgroup analysis derived from source of controls together with genotypic method was preformed.

### Results

Eventually, 28 articles altogether embodying 57 studies were included in this meta-analysis. The results illuminated that LncRNA H19 polymorphisms mentioned above were all irrelevant to cancer susceptibility. Nevertheless, crucial results were found concentrated in population-based control group when subgroup analysis by source of controls were performed in H19 mutation rs2839698 and rs2735971. Meanwhile, in the stratification analysis by genotypic method, apparent cancer risks were discovered by TaqMan method in H19 mutation rs2107425 and rs3024270. Then, trial sequential analysis demonstrated that the results about such associations were firm evidence of effect.

**Funding:** The authors received no specific funding for this work.

**Competing interests:** The authors have declared that no competing interests exist.

## Conclusion

Therefore, this meta-analysis indicated that LncRNA H19 polymorphisms were not associated with the susceptibility to human cancer. However, after the stratification analysis, inconsistent results still existed in different genotypic method and source of control. Thus, more high-quality studies on cancer patients of different factors were needed to confirm these findings.

## Introduction

As a major public health problem in the world, cancer is the second biggest cause of death in the developed countries. 1,762,450 new cancer cases and 606,880 cancer mortalities are predicted to occur in the United States in 2019 [1]. Nevertheless, the pathogeny of malignant tumor still remains vague. Consensus amongst worldwide researchers is that both the environmental and genetic abnormality contribute to the carcinogenesis [2]. The aberration of genetic expression increases the risk of the initiation and progression of cancer [3]. Accumulating studies have been focusing on the repercussions of long non-uncoding RNA (lncRNA) mutation which has a place in the genetic factors mentioned above [4–7].

LncRNA H19 is 2.3 kb in length, located on chromosome 11p15.5 and lacking the open reading frames [8]. The H19 gene, which is maternal imprinted and expressed, plays an irreplaceable role during embryonic phase and decreases in postpartum mature tissues [9]. As we know, LncRNA H19 is considered as a vital factor associated with, cancer susceptibility included, various biological process which impacts the invasion, metastasis, recurrence and poor prognosis of cancer [10]. It might extend the influence upon the development and progression of disease through the regulation of expression on and after transcription of the oncogene and antioncogene [11]. An increasing number of studies have revealed that H19 gene upregulated in almost overall cancer, such as breast cancer, bladder cancer, colorectal cancer, gastric cancer, lung cancer, hepatocellular cnacer, ovarian cancer, pancreatic cancer and so on [4–7, 12].

Single nucleotide polymorphisms (SNPs), a type of genetic mutation, affect the gene expression and function, accordingly causing carcinogenesis [13]. Previous studies have indicated the associations between the risks of cancer and several SNPs (rs2839698 G>A, rs217727 G>A, rs2107425 C>T, rs2735971 A>G and rs3024270 C>G) [2, 14–18]. For instance, Wang et al. conducted a study and found that H19 polymorphism rs217727 might influence the susceptibility to non-small cell lung cancer (NSCLC) [19]. However, another study conducted by Lv et al. showed that H19 polymorphism rs217727 was not associated with overall cancer susceptibility [20]. In that case, though considerable researches have been performed, pooled results seem to be conflicting. Herein, this meta-analysis aimed at deriving a more accurate evaluation in all relevant published studies of the associations between the H19 SNPs and overall cancer susceptibility.

## Materials and methods

We conprehensively retrieved the electronic databases EMBASE, PubMed and Web of Science for all relevant articles published before February 28, 2021, utilizing terms including 'H19 gene', 'polymorphisms' or 'genetic mutation' with 'LncRNA' or 'H19 SNPs', and 'cancer susceptibility' or 'tumor'. Potential eligible studies were collected and integrated by manual work. Additionally, we then removed the duplicate data from different articles.

Meanwhile, the remaining articles were screened by following criteria: (1) Independent case-control or cohort studies; (2) Possessing at least one of H19 polymorphisms (rs2839698 G>A, rs217727 G>A, rs2107425 C>T, rs2735971 A>G and rs3024270 C>G); (3) Availability of subgroup analysis statistical data of both cases and controls; (4) Enrolled patients with cancer diagnosed definitely by histopathological examination, and controls with no history of neoplasms. Correspondingly, the studies enrollment followed these exclusive criteria: (1) Without available data; (2) Without valuable results related to H19; (3) No case-control study.

## Data extraction

The available data from articles after screening were extracted and integrated respectively by two investigators (KP Wang and Z Zhu). Upon the appearance of divergence, a third investigator (YQ Wang) would take intervention and help make a better decision. All extracted data were integrated in an united form, especially with regard to the following information: First author's name, Publication year, Ethnicity, Source of controls, Genotypic method, The number of cases and controls, The number of H19 polymorphisms carriers and non-carriers respectively as well as The results of the Hardy-Weinberg equilibrium (HWE) test.

## Statistical analysis

In the meta-analysis, the pooled odds ratios (ORs) with 95% confidence intervals (CIs) were used to estimate the strength of the associations between the H19 polymorphisms and cancer susceptibility, applying five main genetic comparison models: allele model, homozygous model, heterozygous model, dominant model and recessive model. According to Cochrane Q test and Higgins I [2] statistic, the fixed and random effect model were adopted. I2 < 50% suggested no obvious heterogeneity, in which case fixed effect model should be selected for calculation; only I2 ≧ 50% should the random effect model be selected. Generally, several factors, such as experimental scheme, sex, age, ethnicity, genotypic method and so on, could stimulate the heterogeneity. Therefore, subgroup analysis derived from source of controls and genotypic method was conducted, aiming at investigating the source of heterogeneity. The HWE test was adopted in the control groups to evaluate the gene and genotype frequency, and P value exceeded 0.05, guaranteeing a significant equilibrium. In addition, sensitive analysis was used to examine the stability and reliability of the results through recalculating the pooled ORs following the sequential exclusion of a single study at a time. Meanwhile, we conducted Begg's funnel plots and Egger's linear regression test in order to verify the publication bias among these studies Statistical data was analyzed through Stata statistical software (Version 12.0, Stata Corporation, College Station, TX, USA).

## Trial sequential analysis

The results of the meta-analysis should relate the total number of randomized participants accounting for statistical diversity to avoid type I errors. Thus, trial sequential analysis (TSA) was performed to estimate the required information size, which maintained a 95% confidence interval, a 20% relative risk reduction, an overall type I error of 5% and a statistical test power of 80%. TSA could confirm greater statistical data reliability through modifying the threshold with dispersive data for significance level. We then calculated the required information size and constructed the trial sequential monitoring boundaries. If the blue line (representing the cumulative Z-curve) cross the sloping red line (representing the trial sequential monitoring boundary) or exceed the vertical red line (representing the required information size), a significant result would be reached, and further studies will be unnecessary. On the contrary, either the information size required not being reached or the cumulative Z-curve not crossing the

boundary reveals that additional clinical trials were necessary to reach the sufficient information size for further verification. The TSA software (TSA, version 0.9, 2011; Copenhagen Trial Unit, Copenhagen, Denmark) was adopted in this study.

## Results

### Studies characteristics

Primitively, a total of 262 articles were collected under the guidance of the retrieve strategy above for further screening. Then, 28 articles containing 57 studies met the inclusive criteria, ranging from Feburary 2008 to February 2020 as for publish date [15–19, 21–33]. The flow pathway was shown in **Fig 1** [34–44]. Distribution of the genotypes in the controls was consistent in HWE. The baseline characteristics of all the studies in this meta-analysis were extracted and tabulated in **Table 1**. These studies involved Asians, Caucasians, Africans and Mixed. We separated these studies into two groups, including population-based group and hospital-based group, to help differentiate between various sources of control. Moreover, six genotypic methods altogether were performed in all these studies, such as Taqman, Illumina, PCR-RFLP, Sequenom and so on.

Meanwhile, we calculated the pooled ORs and 95% CIs using five genetic model in order to evaluate the affinity between lncRNA H19 ploymorphisms and cancer susceptibility, results of which were tabulated in **Table 2**. Also, stratification analysis by source of controls and genotypic method was applied to explore the heterogeneity of all studies.

### rs2839698 G>A and cancer susceptibility

Sixteen studies about lncRNA H19 rs2839698 G>A ploymorphism and the susceptibility to cancer consisting 8872 cases and 11,723 controls met the inclusive criteria. The pooled ORs were 1.08 (95% CI: 0.99–1.19) for allele model, 1.08 (95% CI: 0.96–1.21) for dominant model, 1.06 (95% CI: 0.95–1.17) for heterozygote model, 1.16 (95% CI: 0.92–1.42) for homozygote model and 1.13 (95% CI: 0.97–1.31) for recessive model (**Fig 2**). Despite of no positive results, significant association between rs2839698 G>A and cancer susceptibility in population-based controls (allele model: OR = 1.17, 95% CI: 1.04–1.31; homozygote model: OR = 1.41, 95%

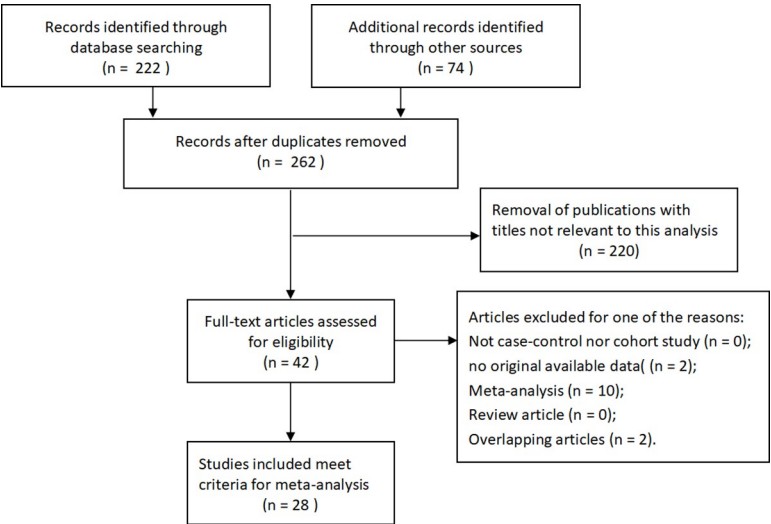

**Fig 1. The flowchart of literature search and selection procedure.**

**Table 1. Characteristics of individual studies included in the meta-analysis.**

| First author | Year | Country/Region | Racial | Source of controls | Case | Control | Genotype distribution | | | | | | Genotyping methods | Test for HWE | | Abbreviation |
|---|---|---|---|---|---|---|---|---|---|---|---|---|---|---|---|---|
| | | | | | | | case | | | control | | | | $\chi^2$ | $P$ | |

**16 Studies for *rs2839698* Poly morphism of the H19 Gene**

| First author | Year | Country/Region | Racial | Source of controls | Case | Control | GG | GA | AA | GG | GA | AA | Genotyping methods | $\chi^2$ | $P$ | Abbreviation |
|---|---|---|---|---|---|---|---|---|---|---|---|---|---|---|---|---|
| Li | 2016 | China | Asian | PB | 1147 | 1203 | 583 | 462 | 102 | 666 | 462 | 75 | TaqMan | 0.582 | 0.446 | CRC |
| Hua | 2016 | China | Asian | HB | 1049 | 1397 | 552 | 418 | 79 | 729 | 565 | 103 | TaqMan | 0.000 | 0.991 | BC |
| Gong | 2016 | China | Asian | HB | 496 | 206 | 237 | 220 | 39 | 99 | 80 | 27 | TaqMan | 1.515 | 0.218 | LC |
| Yang | 2015 | China | Asian | HB | 500 | 500 | 250 | 195 | 55 | 284 | 178 | 38 | TaqMan | 3.206 | 0.073 | GC |
| Verhaegh | 2008 | Netherlands | Caucasian | PB | 177 | 204 | 54 | 74 | 49 | 52 | 109 | 43 | PCR-RFLP | 4.713 | 0.030 | BC |
| Yang | 2018 | China | Asian | HB | 472 | 472 | 215 | 211 | 40 | 245 | 185 | 32 | KASP | 1.372 | 0.241 | HCC |
| Guo | 2017 | China | Asian | HB | 362 | 741 | 133 | 171 | 58 | 244 | 377 | 120 | Illumina | 0.060 | 0.806 | OSCC |
| He | 2017 | China | Asian | HB | 193 | 383 | 83 | 98 | 12 | 178 | 175 | 30 | TaqMan | 5.876 | 0.015 | Osteosarcoma |
| Cui | 2018 | China | Asian | HB | 1491 | 1677 | 801 | 568 | 122 | 875 | 673 | 129 | TaqMan | 2.238 | 0.135 | BCa |
| Mohammad | 2019 | Iran | Asian | HB | 111 | 130 | 15 | 57 | 39 | 53 | 55 | 22 | 4P-ARMSPCR | 0.665 | 0.415 | BCa |
| Wu | 2019 | China | Asian | HB | 359 | 1190 | 140 | 178 | 41 | 532 | 524 | 134 | TaqMan | 1.927 | 0.165 | HCC |
| Yang | 2019 | China | Asian | HB | 431 | 431 | 206 | 170 | 55 | 192 | 184 | 55 | PCR | 4.376 | 0.036 | BC |
| Wang | 2019 | China | Asian | HB | 563 | 1532 | 277 | 225 | 61 | 712 | 645 | 175 | TaqMan | 2.217 | 0.136 | LC |
| Lin | 2017 | China | Asian | HB | 1005 | 1020 | 452 | 440 | 113 | 484 | 432 | 104 | PCR | 0.144 | 0.705 | BCa |
| Yu | 2020 | China | Asian | HB | 315 | 441 | 134 | 140 | 40 | 154 | 211 | 74 | PCR | 0.132 | 0.716 | CRC |
| Zhang | 2020 | China | Asian | HB | 201 | 196 | 70 | 93 | 38 | 92 | 88 | 16 | Sequenom MassARRAY | 0.514 | 0.473 | OC |

**17 Studies for *rs217727 G* Poly morphism of the H19 Gene**

| First author | Year | Country/Region | Racial | Source of controls | Case | Control | GG | GA | AA | GG | GA | AA | Genotyping methods | $\chi^2$ | $P$ | Abbreviation |
|---|---|---|---|---|---|---|---|---|---|---|---|---|---|---|---|---|
| Hua | 2016 | China | Asian | HB | 1046 | 1394 | 431 | 467 | 148 | 573 | 665 | 156 | TaqMan | 1.397 | 0.237 | BC |
| Li | 2016 | China | Asian | PB | 1147 | 1203 | 480 | 514 | 153 | 456 | 570 | 177 | TaqMan | 0.686 | 0.407 | CRC |
| Xia | 2016 | China | Asian | PB | 464 | 467 | 160 | 156 | 148 | 139 | 212 | 116 | CRS-RFLP | 0.490 | 0.000 | BCa |
| Yang | 2015 | China | Asian | HB | 500 | 500 | 160 | 252 | 88 | 193 | 244 | 63 | TaqMan | 0.431 | 0.512 | GC |
| Verhaegh | 2008 | Netherlands | Caucasian | PB | 177 | 204 | 114 | 59 | 4 | 115 | 80 | 9 | PCR-RFLP | 1.314 | 0.252 | BC |
| Yin | 2018 | China | Asian | PB | 556 | 395 | 204 | 264 | 88 | 165 | 172 | 58 | Illumina | 0.028 | 0.866 | LC |
| Guo | 2017 | China | Asian | HB | 362 | 740 | 101 | 181 | 80 | 255 | 348 | 137 | Illumina | 0.004 | 0.949 | OSCC |
| He | 2017 | China | Asian | HB | 193 | 383 | 79 | 102 | 12 | 195 | 165 | 23 | TaqMan | 7.862 | 0.005 | Osteosarcoma |
| Abdollahzadeh | 2018 | Iran | Asian | HB | 150 | 100 | 116 | 29 | 5 | 86 | 14 | 0 | PCR-RFLP | 3.167 | 0.075 | BCa |
| Cui | 2018 | China | Asian | HB | 1488 | 1675 | 611 | 692 | 185 | 685 | 773 | 217 | TaqMan | 0.257 | 0.612 | BCa |
| Hu | 2017 | China | Asian | HB | 416 | 416 | 133 | 200 | 83 | 128 | 196 | 92 | TaqMan | 0.247 | 0.619 | PC |
| Mohammad | 2019 | Iran | Asian | HB | 111 | 130 | 79 | 30 | 2 | 64 | 54 | 12 | 4P-ARMSPCR | 0.195 | 0.659 | BCa |
| Wu | 2019 | China | Asian | HB | 359 | 1190 | 154 | 170 | 35 | 495 | 539 | 156 | TaqMan | 1.470 | 0.225 | HCC |
| Yang | 2019 | China | Asian | HB | 431 | 431 | 185 | 202 | 44 | 191 | 188 | 52 | PCR | 1.065 | 0.302 | BC |
| Wang | 2019 | China | Asian | HB | 564 | 1535 | 162 | 277 | 125 | 493 | 751 | 291 | TaqMan | 0.103 | 0.749 | LC |
| Li | 2019 | China | Asian | HB | 200 | 200 | 51 | 140 | 9 | 84 | 90 | 26 | TaqMan | 43.168 | 0.000 | BC |
| Lin | 2017 | China | Asian | HB | 1005 | 1020 | 403 | 471 | 131 | 465 | 450 | 105 | PCR | 0.131 | 0.718 | BCa |
| Li | 2018 | China | Asian | HB | 555 | 618 | 210 | 250 | 95 | 246 | 305 | 97 | TaqMan | 1.911 | 0.167 | LC |

**10 Studies for *rs2107425* Poly morphism of the H19 Gene**

| First author | Year | Country/Region | Racial | Source of controls | Case | Control | CC | CT | TT | CC | CT | TT | Genotyping methods | $\chi^2$ | $P$ | Abbreviation |
|---|---|---|---|---|---|---|---|---|---|---|---|---|---|---|---|---|
| Verhaegh | 2008 | Netherlands | Caucasian | PB | 177 | 204 | 92 | 65 | 20 | 89 | 96 | 19 | PCR-RFLP | 2.545 | 0.111 | BC |
| Song | 2009 | Mixed | Caucasian | PB | 5366 | 8538 | 2619 | 2192 | 555 | 4029 | 3667 | 842 | TaqMan, Sequenom MassArray | 9.082 | 0.003 | OC |
| Quaye | 2009 | Mixed | Caucasian | PB | 1460 | 2463 | 767 | 544 | 149 | 1118 | 1098 | 247 | TaqMan | 12.392 | 0.000 | OC |

*(Continued)*

**Table 1.** (Continued)

| First author | Year | Country/ Region | Racial | Source of controls | Case | Control | Genotype distribution | | | | | | Genotyping methods | Test for HWE | | Abbreviation |
|---|---|---|---|---|---|---|---|---|---|---|---|---|---|---|---|---|
| | | | | | | | case | | | control | | | | $\chi^2$ | P | |
| Barnholtz-Sloan 11 | 2014 | USA | Caucasian | PB | 1225 | 1118 | 604 | 516 | 105 | 521 | 478 | 119 | Illumina | 0.124 | 0.725 | BCa |
| Barnholtz Sloan 12 | 2014 | USA | Africa | PB | 737 | 658 | 161 | 390 | 186 | 170 | 339 | 149 | Illumina | 2.615 | 0.106 | BCa |
| Butt | 2012 | Sweden | Caucasian | PB | 678 | 1355 | 360 | 250 | 68 | 637 | 573 | 145 | Sequenom | 6.067 | 0.014 | BCa |
| Gong | 2016 | China | Asian | HB | 479 | 203 | 181 | 235 | 63 | 79 | 96 | 28 | TaqMan | 0.953 | 0.329 | LC |
| Yin | 2018 | China | Asian | PB | 556 | 395 | 161 | 266 | 129 | 140 | 185 | 70 | Illumina | 0.889 | 0.346 | LC |
| Wu | 2019 | China | Asian | HB | 359 | 1190 | 134 | 185 | 40 | 422 | 560 | 208 | TaqMan | 4.072 | 0.044 | HCC |
| Yang | 2019 | China | Asian | HB | 431 | 431 | 152 | 213 | 66 | 171 | 190 | 70 | PCR | 0.372 | 0.542 | BC |
| Yin | 2018 | China | Asian | HB | 556 | 395 | 161 | 266 | 129 | 140 | 185 | 70 | Illumina | 0.889 | 0.346 | LC |
| **6 Studies for *rs2735971* Poly morphism of the H19 Gene** | | | | | | | | | | | | | | | | |
| | | | | | | | AA | AG | GG | AA | AG | GG | | | | |
| Hua | 2016 | China | Asian | HB | 1049 | 1396 | 43 | 302 | 704 | 46 | 422 | 928 | TaqMan | 2.128 | 0.145 | BC |
| Li | 2016 | China | Asian | PB | 1147 | 1203 | 773 | 334 | 40 | 765 | 398 | 40 | TaqMan | 0.278 | 0.598 | CRC |
| Yang | 2018 | China | Asian | HB | 472 | 472 | 12 | 126 | 327 | 13 | 139 | 313 | KASP | 0.001 | 0.974 | HCC |
| Guo | 2017 | China | Asian | HB | 461 | 739 | 129 | 141 | 191 | 80 | 308 | 351 | Illumina | 65.528 | 0.000 | OSCC |
| He | 2017 | China | Asian | HB | 193 | 383 | 11 | 94 | 88 | 32 | 182 | 169 | TaqMan | 4.848 | 0.028 | Osteosarcoma |
| Li | 2019 | China | Asian | HB | 200 | 200 | 10 | 62 | 128 | 4 | 70 | 126 | TaqMan | 0.479 | 0.489 | BC |
| **8 Studies for *rs3024270* Poly morphism of the H19 Gene** | | | | | | | | | | | | | | | | |
| | | | | | | | CC | GC | GG | CC | GC | GG | | | | |
| Hua | 2016 | China | Asian | HB | 1047 | 1395 | 174 | 527 | 346 | 260 | 688 | 447 | TaqMan | 1.254 | 0.263 | BC |
| Li | 2016 | China | Asian | PB | 1147 | 1203 | 385 | 527 | 235 | 420 | 582 | 201 | TaqMan | 4.860 | 0.027 | CRC |
| Yang | 2018 | China | Asian | HB | 472 | 472 | 95 | 225 | 151 | 81 | 215 | 170 | KASP | 0.449 | 0.503 | HCC |
| Guo | 2017 | China | Asian | HB | 362 | 740 | 75 | 183 | 104 | 145 | 350 | 245 | Illumina | 0.112 | 0.738 | OSCC |
| He | 2017 | China | Asian | HB | 193 | 383 | 17 | 91 | 85 | 31 | 179 | 173 | TaqMan | 1.134 | 0.287 | OSC |
| Wu | 2019 | China | Asian | HB | 359 | 1190 | 87 | 187 | 85 | 334 | 593 | 263 | TaqMan | 0.628 | 0.428 | HCC |
| Yang | 2019 | China | Asian | HB | 431 | 431 | 114 | 210 | 107 | 120 | 208 | 103 | PCR | 0.275 | 0.600 | BC |
| Li | 2019 | China | Asian | HB | 200 | 200 | 16 | 101 | 83 | 22 | 97 | 81 | TaqMan | 3.791 | 0.052 | BC |

**Abbreviation:** SOC: source of control; HB: hospital-based; HWE: Hardy–Weinberg equilibrium; BC: Bladder cancer; BCa: Breast cancer; LC: Lung cancer; HCC: Hepatocellular cancer; OSCC: Oral squamous cell carcinoma; OSC:Osteosarcoma; PC: Pancreatic cancer; CRC: Colorectal cancer; GC: Gastric cancer; OC: Ovarian cancer.

CI: 1.04–1.91; recessive model: OR = 1.46, 95% CI: 1.13–1.89) was observed in the stratification analysis by source of control. In addition, no significant results were detected in the subgroup analysis by genotypic method.

## rs217727 G>A and cancer susceptibility

In this meta-analysis, 17 Studies focusing on rs217727 G>A polymorphism and cancer susceptibility included 8678 cases and 11,207 controls. No significant association was indicated through the pooled risk estimation under allele model (OR = 1.04, 95% CI = 0.96–1.13), dominant model (OR = 1.07, 95% CI = 0.95–1.21), heterozygous model (OR = 1.07, 95% CI = 0.94–1.21), homozygous model (OR = 1.06, 95% CI = 0.90–1.24) and recessive model (OR = 1.06, 95% CI = 0.79–1.42) (Fig 3). While no significant results were observed in subgroup analysis

**Table 2. Meta-analysis results for the included studies of the association between LncRNA H19 polymorphisms and risk of cancer.**

| Variables | No. of studies | Allele model | | | Dominant model | | | Heterozygous model | | | Homozygous model | | | Recessive model | | |
|---|---|---|---|---|---|---|---|---|---|---|---|---|---|---|---|---|
| | | OR (95% CI) | P values | I-squared (%) | OR (95% CI) | P values | I-squared (%) | OR (95% CI) | P values | I-squared (%) | OR (95% CI) | P values | I-squared (%) | OR (95% CI) | P values | I-squared (%) |
| **1. rs2839698 G>A** | | A vs G | | | (GA+AA) vs GG | | | GA vs GG | | | AA vs GG | | | AA vs (GA+GG) | | |
| All | 16 | 1.08(0.99,1.19) | <0.001 | 75.3 | 1.08(0.96,1.21) | <0.001 | 71.7 | 1.06(0.95,1.17) | 0.001 | 63.3 | 1.16(0.92,1.42) | <0.001 | 71.7 | 1.13(0.97,1.31) | 0.002 | 58.1 |
| Source of control | | | | | | | | | | | | | | | | |
| PB | 2 | **1.17 (1.04,1.31)** | 0.347 | <0.1 | 1.02(0.68,1.54) | 0.076 | 68.2 | 0.91(0.53,1.55) | 0.032 | 78.2 | **1.41 (1.04,1.91)** | 0.290 | 10.9 | **1.46 (1.13,1.89)** | 0.934 | <0.1 |
| HB | 14 | 1.08(0.97,1.19) | <0.001 | 76.9 | 1.09(0.96,1.24) | <0.001 | 73.2 | 1.07(0.95,1.20) | 0.001 | 64.0 | 1.14(0.92,1.42) | <0.001 | 73.1 | 1.08(0.92,1.28) | 0.004 | 57.9 |
| Method of genotype | | | | | | | | | | | | | | | | |
| TaqMan | 8 | 1.05(0.96,1.14) | 0.024 | 56.5 | 1.07(0.96,1.18) | 0.052 | 49.9 | 1.06(0.96,1.17) | 0.108 | 40.6 | 1.08(0.89,1.32) | 0.038 | 52.9 | 1.05(0.88,1.26) | 0.056 | 49.0 |
| non-TaqMan | 8 | 1.15(0.95,1.39) | <0.001 | 84.2 | 1.12(0.87,1.45) | <0.001 | 82.1 | 1.05(0.84,1.33) | <0.001 | 75.9 | 1.34(0.91,1.96) | <0.001 | 81.5 | 1.29(0.95,1.76) | 0.053 | 57.1 |
| **2. rs217727 G>A** | | A vs G | | | (GA+AA) vs GG | | | GA vs GG | | | AA vs GG | | | AA vs (GA+GG) | | |
| All | 17 | 1.04(0.96,1.13) | <0.001 | 69.9 | 1.07(0.95,1.21) | <0.001 | 72.4 | 1.07(0.94,1.21) | <0.001 | 71.9 | 1.06(0.90,1.24) | 0.001 | 59.2 | 1.03(0.89,1.19) | 0.001 | 59.7 |
| Source of control | | | | | | | | | | | | | | | | |
| PB | 4 | 0.97(0.83,1.13) | 0.051 | 61.4 | 0.90(0.72,1.11) | 0.048 | 62.1 | 0.85(0.66,1.11) | 0.014 | 71.6 | 0.97(0.75,1.26) | 0.154 | 42.9 | 1.06(0.79,1.42) | 0.056 | 60.4 |
| HB | 13 | 1.07(0.97,1.18) | <0.001 | 70.4 | **1.14 (0.99,1.30)** | <0.001 | 70.1 | **1.15 (1.00,1.31)** | 0.001 | 63.3 | 1.09(0.90,1.31) | 0.002 | 62.2 | 1.01(0.85,1.20) | 0.001 | 62.7 |
| Method of genotype | | | | | | | | | | | | | | | | |
| TaqMan | 9 | 1.04(0.95,1.14) | 0.009 | 60.8 | 1.11(0.96,1.28) | 0.001 | 70.4 | 1.12(0.96,1.31) | <0.001 | 71.7 | 1.02(0.84,1.22) | 0.015 | 57.8 | 0.97(0.81,1.17) | 0.006 | 62.6 |
| non-TaqMan | 8 | 1.02(0.86,1.21) | <0.001 | 76.9 | 1.01(0.80,1.26) | <0.001 | 77.0 | 0.98(0.77,1.24) | <0.001 | 75.5 | 1.11(0.84,1.47) | 0.018 | 58.8 | 1.12(0.88,1.41) | 0.044 | 51.6 |
| **3. rs2107425 C>T** | | T vs C | | | (CT+TT) vs CC | | | CT vs CC | | | TT vs CC | | | TT vs (CT+CC) | | |
| All | 10 | 0.96(0.89,1.04) | 0.001 | 68.5 | 0.95(0.85,1.06) | <0.001 | 71.5 | 0.95(0.84,1.07) | <0.001 | 71.6 | 0.97(0.83,1.13) | 0.010 | 58.6 | 0.98(0.87,1.12) | 0.035 | 50.1 |
| Source of control | | | | | | | | | | | | | | | | |
| PB | 7 | 0.96(0.88,1.06) | <0.001 | 75.8 | 0.92(0.80,1.06) | <0.001 | 77.8 | 0.90(0.78,1.03) | <0.001 | 75.5 | 1.01(0.85,1.19) | 0.016 | 61.5 | 1.04(0.93,1.17) | 0.190 | 31.2 |
| HB | 3 | 0.96(0.82,1.13) | 0.154 | 46.6 | 1.04(0.88,1.23) | 0.360 | 2.1 | 1.12(0.94,1.32) | 0.602 | <0.1 | 0.85(0.59,1.22) | 0.114 | 54.0 | 0.79(0.58,1.09) | 0.150 | 47.3 |
| Method of genotype | | | | | | | | | | | | | | | | |
| TaqMan | 3 | **0.86(0.80,0.94)** | 0.396 | <0.1 | 0.86(0.71,1.05) | 0.098 | 57.0 | 0.90(0.68,1.21) | 0.013 | 76.9 | 0.81(0.62,1.04) | 0.206 | 36.7 | 0.84(0.59,1.20) | 0.038 | 69.3 |
| non-TaqMan | 7 | 1.00(0.91,1.10) | 0.004 | 68.9 | 0.99(0.86,1.13) | 0.002 | 70.6 | 0.97(0.85,1.11) | 0.006 | 66.9 | 1.04(0.87,1.24) | 0.028 | 57.6 | 1.05(0.96,1.15) | 0.124 | 42.2 |
| **4. rs2735971 A>G** | | G vs A | | | (AG+GG) vs AA | | | AG vs AA | | | GG vs AA | | | GG vs (AG+AA) | | |
| All | 6 | 0.91(0.75,1.11) | <0.001 | 81.9 | 0.72(0.44,1.17) | <0.001 | 86.8 | 0.68(0.41,1.13) | <0.001 | 86.3 | 0.76(0.45,1.29) | <0.001 | 81.5 | 0.99(0.89,1.11) | 0.247 | 26.2 |
| Source of control | | | | | | | | | | | | | | | | |
| PB | 1 | 0.89(0.77,1.03) | - | - | **0.85 (0.71,1.00)** | - | - | **0.83 (0.70,0.99)** | - | - | 0.99(0.63,1.55) | - | - | 1.05(0.67,1.64) | - | - |
| HB | 5 | 0.92(0.72,1.19) | <0.001 | 85.5 | 0.69(0.35,1.34) | <0.001 | 85.0 | 0.65(0.33,1.29) | <0.001 | 84.4 | 0.71(0.38,1.34) | <0.001 | 82.1 | 0.99(0.88,1.11) | 0.247 | 26.2 |
| **5. rs3024270 C>G** | | G vs C | | | (CG+GG) vs CC | | | CG vs CC | | | GG vs CC | | | GG vs (CG+CC) | | |
| All | 8 | 1.03(0.98,1.10) | 0.237 | 24.1 | 1.07(0.97,1.18) | 0.630 | <0.1 | 1.05(0.95,1.17) | 0.811 | <0.1 | 1.09(0.97,1.23) | 0.218 | 21.4 | 1.03(0.94,1.13) | 0.177 | 31.5 |
| Source of control | | | | | | | | | | | | | | | | |
| PB | 1 | 1.11(0.99,1.25) | - | - | 1.06(0.90,1.26) | - | - | 0.99(0.82,1.18) | - | - | **1.28 (1.01,1.61)** | - | - | 1.28(1.04,1.58) | - | - |
| HB | 7 | 1.01(0.95,1.08) | 0.293 | 17.9 | 1.07(0.95,1.20) | 0.513 | <0.1 | 1.09(0.96,1.23) | 0.809 | <0.1 | 1.04(0.90,1.19) | 0.297 | 17.5 | 0.98(0.88,1.08) | 0.563 | <0.1 |
| Method of genotype | | | | | | | | | | | | | | | | |
| TaqMan | 5 | **1.08 (1.01,1.16)** | 0.873 | <0.1 | **1.12 (1.00,1.26)** | 0.788 | <0.1 | 1.08(0.96,1.22) | 0.622 | <0.1 | **1.21 (1.05,1.39)** | 0.851 | <0.1 | 1.10(0.99,1.23) | 0.534 | <0.1 |
| non-TaqMan | 3 | 0.93(0.84,1.03) | 0.309 | 14.9 | 0.95(0.79,1.14) | 0.532 | <0.1 | 0.99(0.82,1.20) | 0.763 | <0.1 | 0.88(0.71,1.08) | 0.353 | 4.1 | 0.88(0.74,1.03) | 0.404 | <0.1 |

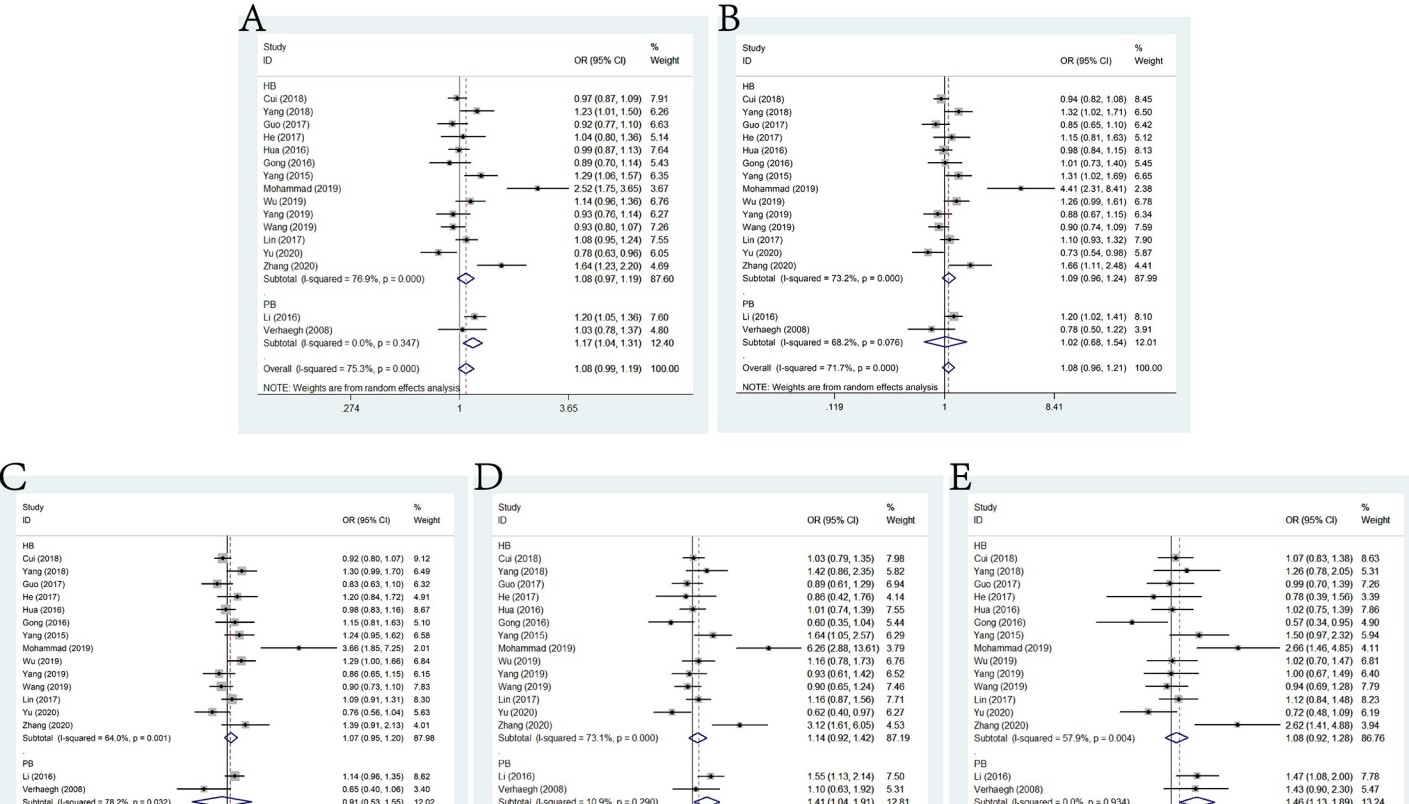

**Fig 2. Forest plot of the association between H19 polymorphism rs2839698 G>A and cancer susceptibility.** A: allele model; B: dominant model; C: heterozygote model; D: homozygote model; E: recessive model.

by genotypic method, positive results were found in hospital-based controls (heterozygous model: OR = 1.15, 95% CI: 1.00–1.31).

## rs2107425 C>T and cancer susceptibility

A total of 10 studies embodying 11,468 cases and 16,555 controls were investigated to LncRNA H19 polymorphic variants rs2107425 C>T and the susceptibility to cancer. Analogously, no significant association between rs2107425 C>T polymorphism and cancer risk was shown in the meta-analysis according to the pooled ORs of these studies under allele model (OR = 0.96, 95% CI = 0.89–1.04), dominant model (OR = 0.95, 95% CI = 0.85–1.06), heterozygous model (OR = 0.95, 95% CI = 0.84–1.07), homozygous model (OR = 0.97, 95% CI = 0.83–1.13) and recessive model (OR = 0.98, 95% CI = 0.87–1.12) (Fig 4). Nevertheless, as to the stratification analysis of genotypic method, the result was significant only in TaqMan (allele model: OR = 0.86, 95% CI = 0.80–0.94), while no significant results was detected in subgroup of source of control.

## rs2735971 A>G and cancer susceptibility

The present meta-analysis enrolled 3,522 cases and 4,393 controls from a sum of six studies on rs2735971 A>G polymorphism and cancer susceptibility. No significant association was observed among these studies under all the genetic models (allele model (OR = 0.91, 95%

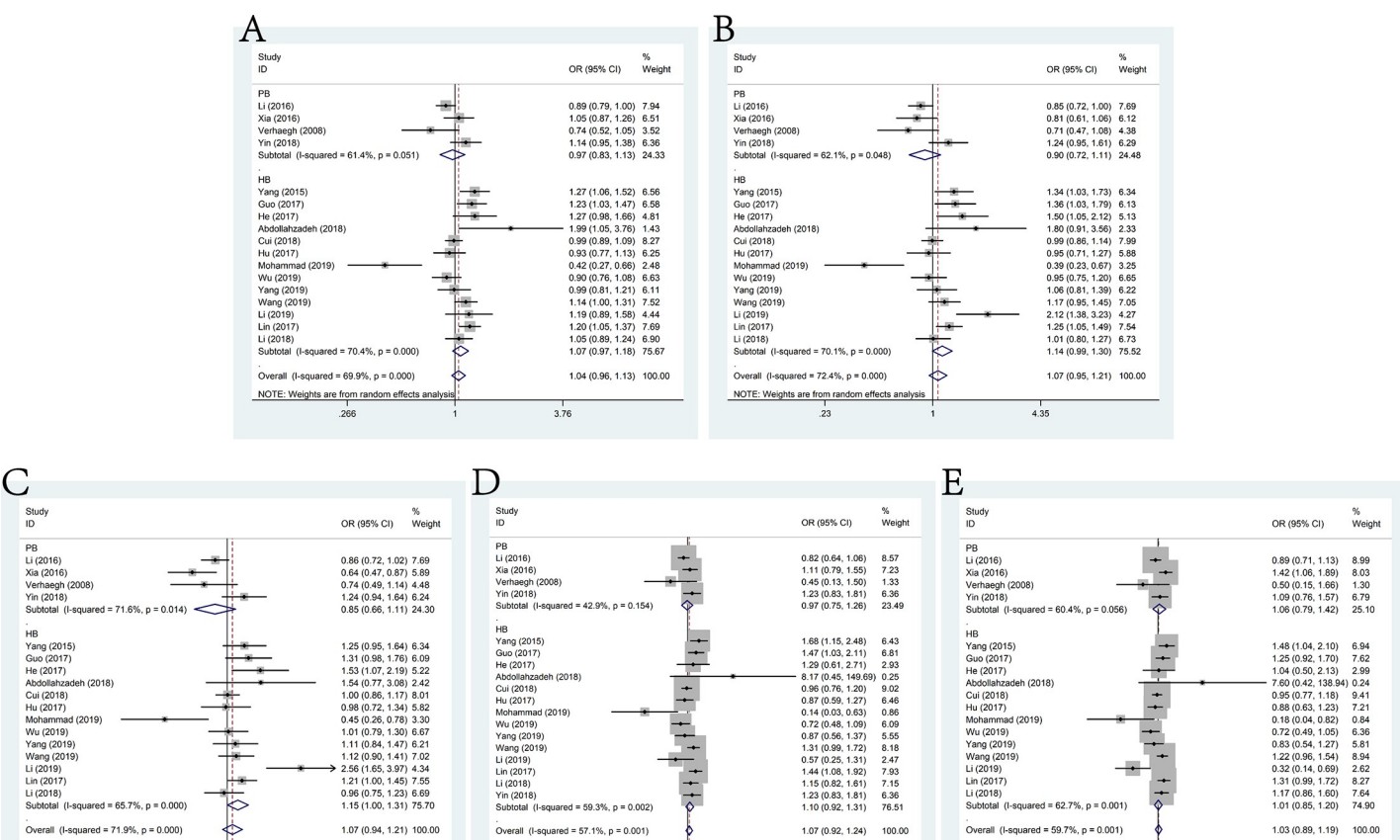

**Fig 3. Forest plot of the association between H19 polymorphism rs217727 G>A and cancer susceptibility.** A: allele model; B: dominant model; C: heterozygote model; D: homozygote model; E: recessive model.

CI = 0.75–1.11), dominant model (OR = 0.72, 95% CI = 0.44–1.17), heterozygous model (OR = 0.68, 95% CI = 0.41–1.13), homozygous model (OR = 0.76, 95% CI = 0.45–1.29) and recessive model (OR = 0.99, 95% CI = 0.89–1.11)) (Fig 5). Additionally, the results of stratified analysis by genotypic method were not positive. By contrast, in subgroup analysis by source of control, feebly positive results were shown in population-based controls(dominant model: OR = 0.85, 95% CI = 0.71–1.00; heterozygous model: OR = 083, 95% CI = 0.70–0.99).

## rs3024270 C>G and cancer susceptibility

No significant association existed between rs3024270 mutation and cancer susceptibility as shown by the pooled risks of 8 relevant studies consisting 4,211 cases and 6,014 controls under allele model (OR = 1.03, 95% CI = 0.98–1.10), dominant model(OR = 1.07, 95% CI = 0.97–1.18), heterozygous model(OR = 1.05, 95% CI = 0.95–1.17), homozygous model (OR = 1.09, 95% CI = 0.97–1.23) and recessive model(OR = 1.03, 95% CI = 0.94–1.13) (Fig 6). However, stratification analysis by source of control indicated significant association with cancer susceptibility in the population-based control group (homozygous model: OR = 1.28, 95% CI = 1.01–1.61). Furthermore, the results of analysis stratified by genotypic method were more significant while using Taq-Man than non-TaqMan methods (allele model: OR = 1.08, 95% CI = 1.01–1.16, dominant model: OR = 1.12, 95% CI = 1.00–1.26 and homozygous model: OR = 1.21, 95% CI = 1.05–1.39).

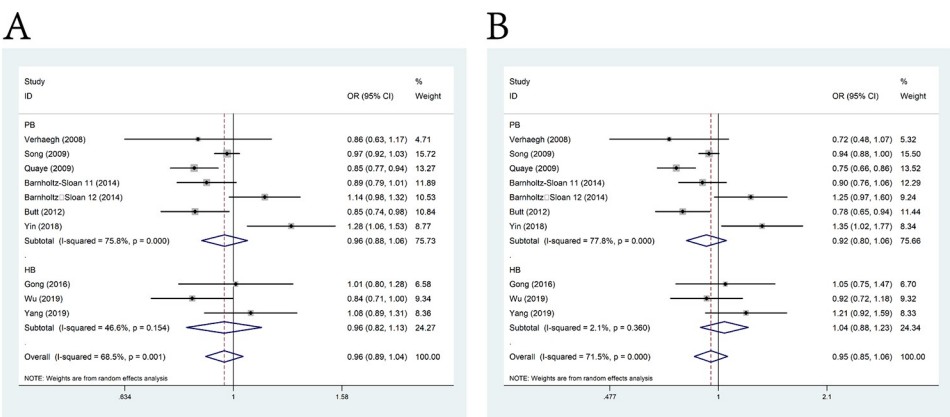

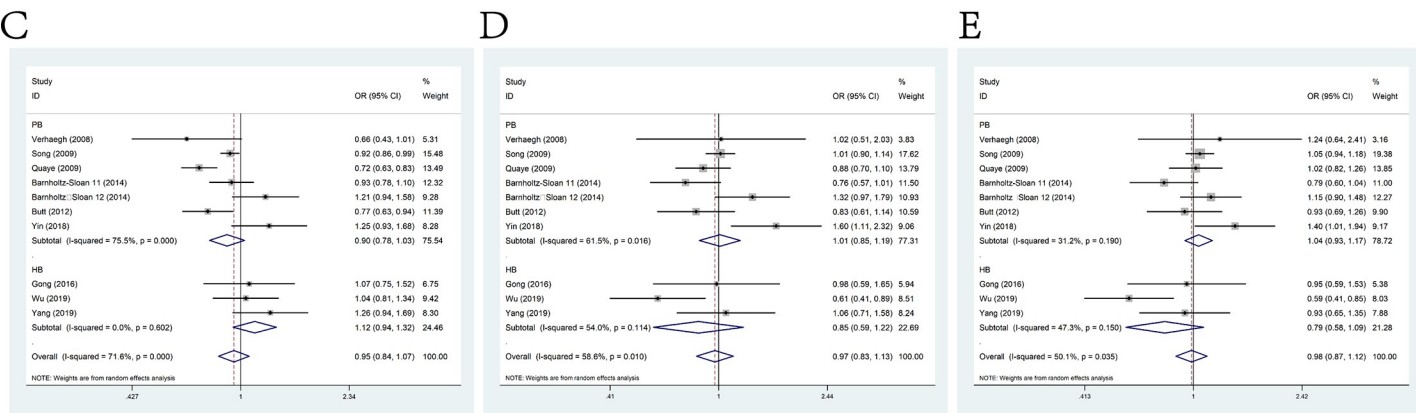

**Fig 4. Forest plot of the association between H19 polymorphism rs2107425 C>T and cancer susceptibility.** A: allele model; B: dominant model; C: heterozygote model; D: homozygote model; E: recessive model.

## Sensitivity analysis

Sensitivity analysis was carried out by removing single eligible study sequentially to detect individual study's influence on the pooled results. According to the results, no single study was found affect the pooled OR in the allele model, suggesting a statistically robust results (**Fig 7**).

## Publication bias

The Begg's funnel plot and Egger's test were utilized in the selected literature. With the symmetrical shapes of funnel plots shown in **Fig 8**, the absence of publication bias could be testified in the allele model (rs2839698: Begg's Test $P = 0.207$ Egger's test $P = 0.169$, rs217727: Begg's Test $P = 0.805$ Egger's test $P = 0.943$, rs2107425: Begg's Test $P = 0.421$ Egger's test $P = 0.835$, rs2735971: Begg's Test $P = 0.851$ Egger's test $P = 0.593$ and rs3024270: Begg's Test $P = 0.322$ Egger's test $P = 0.305$).

## Trial sequential analysis results

In this meta-analysis, **Fig 9** showed that the cumulative Z-curve of all the H19 mutations investigated either crossed the trial sequential monitoring boundary or exceeded the required information size, indicating that the results about the associations between LncRNA H19 polymorphic variants (rs2839698 G>A, rs217727 G>A, rs2107425 C>T, rs2735971 A>G and rs3024270 C>G) and the susceptibility to cancer were firm evidence of effect.

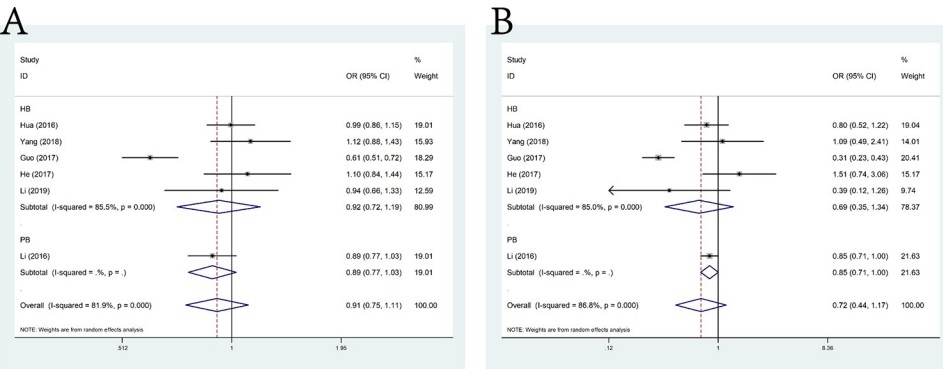

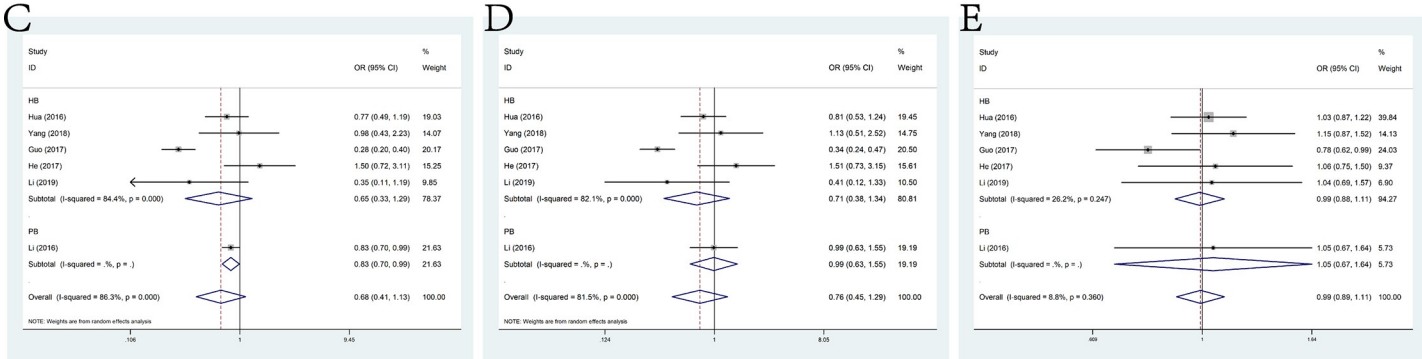

**Fig 5. Forest plot of the association between H19 polymorphism rs2735971 A>G and cancer susceptibility.** A: allele model; B: dominant model; C: heterozygote model; D: homozygote model; E: recessive model.

## Discussion

An increasing number of studies have been focusing on the mutation of H19 when it comes to the genesis and development of various cancer [45]. As a long non-coding RNA, H19 lacks the open reading frame to translate protein whose end product is RNA sequence and can participate in downstream RNA regulatory [21, 46]. LncRNA H19 is an imprinted gene the aberrant expression of which is associated with cancer susceptibility [22]. In this meta-analysis, SNPs (rs2839698 G>A, rs217727 G>A, rs2107425 C>T, rs2735971 A>G and rs3024270 C>G) were included to investigate the relationship between these polymorphisms and the risk of cancer.

Previously, several meta-analysis on aberration of H19 have been conducted thanks to the identification of numerous LncRNAs [23, 24]. Though the results were contrary to many studies, a meta-analysis performed by Lv. revealed that rs217727 were uncorrelated to overall cancer risk [20]. It might account for the lack of the interactive microRNAs (miRNAs) which could influence the regulation and modification of lncRNAs SNPs directly [20, 47]. In that case, the position where gene structural changes caused by the polymorphism might differ from where the gene binds with elements such as miRNAs that regulate lncRNA expression, thus indicating no significant association with overall cancer risk. Also, another meta-analysis conducted by Li et al. inspired us on the possible reason [48]. The various cancer location and patient ethnicity might accounting for the discrepancies among the studies examined. In these studies, a small sample size and controversial results caused by the former factor might make

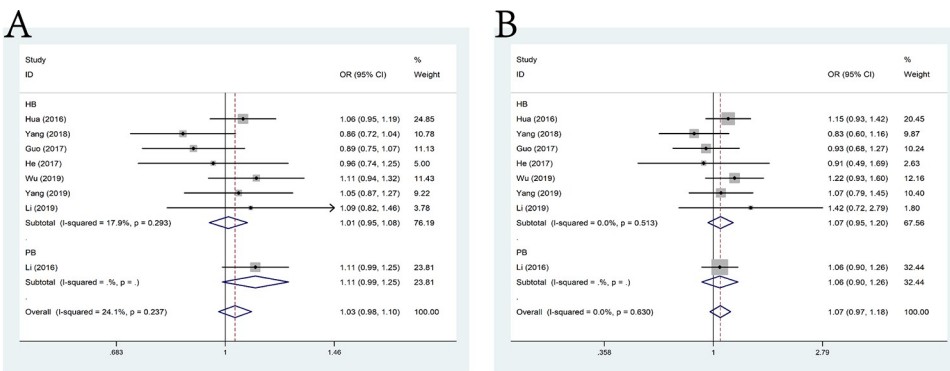

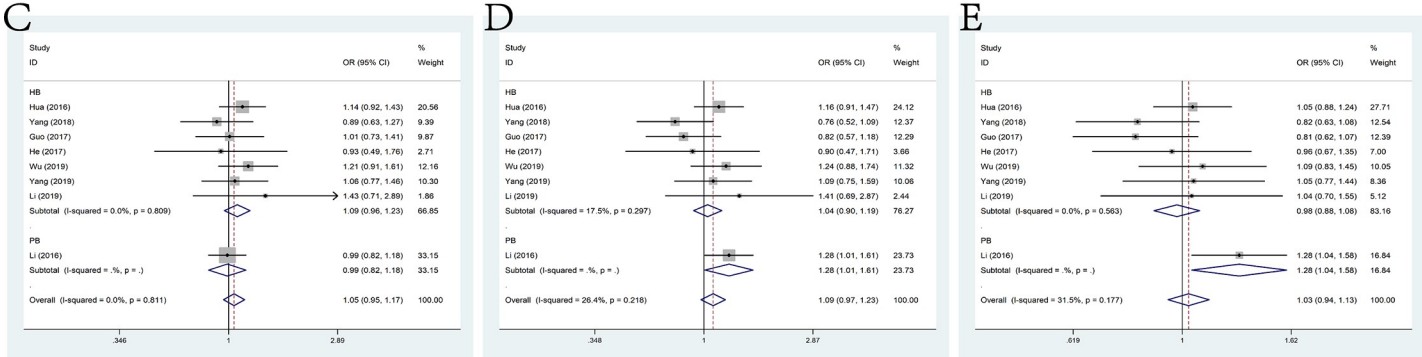

**Fig 6. Forest plot of the association between H19 polymorphism rs3024270 C>G and cancer susceptibility.** A: allele model; B: dominant model; C: heterozygote model; D: homozygote model; E: recessive model.

these analysis relatively unreliable. Herein, we conducted this meta-analysis with the largest sample capacity and the most up-to-date studies and data, comprehensively analyzing all literatures to study such association. According to quantitative synthesis results, all the mutation mentioned above were found no significant association.

When stratified by source of control, significant association was found in the population-based control group between rs2839698, rs2735971 and rs3024270 polymorphisms and the susceptibility to cancer, whereas significant results in hospital-based control group were only found in SNP rs217727. Lack of the representativeness might account for the phenomena. Moreover, in the subgroup analysis by genotypic method, significant results were also found between the risk of cancer and rs2107425, rs3024270 polymorphisms adopting TaqMan method for genotyping, whereas similar results were not found while using other genotypic methods. The possible reason might be that different genotypic methods lead to different statistical results owing to their relative merits. The merits of TaqMan are the lower probability of PCR pollution due to that the reaction happens in the PCR process, avoiding separation and elution process [49].

TSA, as an statistical tool, is similar to interim analysis in a single trial, where trial monitoring boundaries are drawn for each outcome whether to continue additional trials to assess for evidence while a P value is small enough to show the projected effect or for futility [50]. The association shown in the results of this meta-analysis could be unreliable accounting for limited data. Therefore, TSA was adopted in order to diminish the probability of type I error and verify whether the evidence of our results was adequate or not. The results about the

A

B

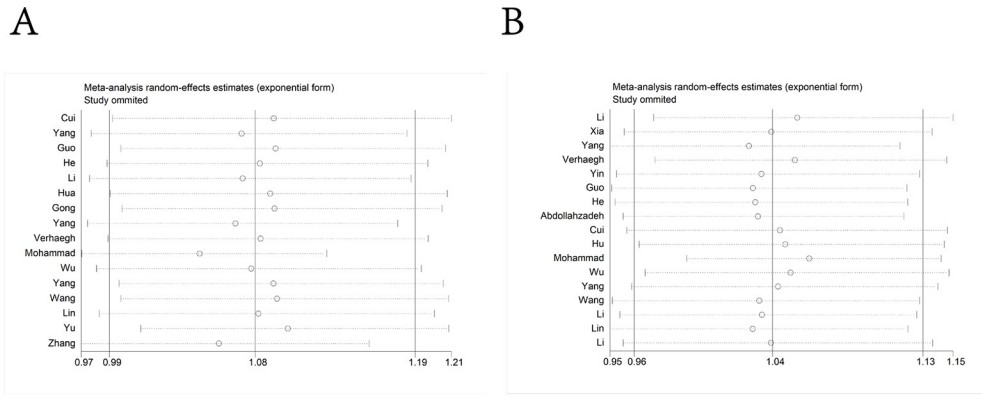

C

D

E

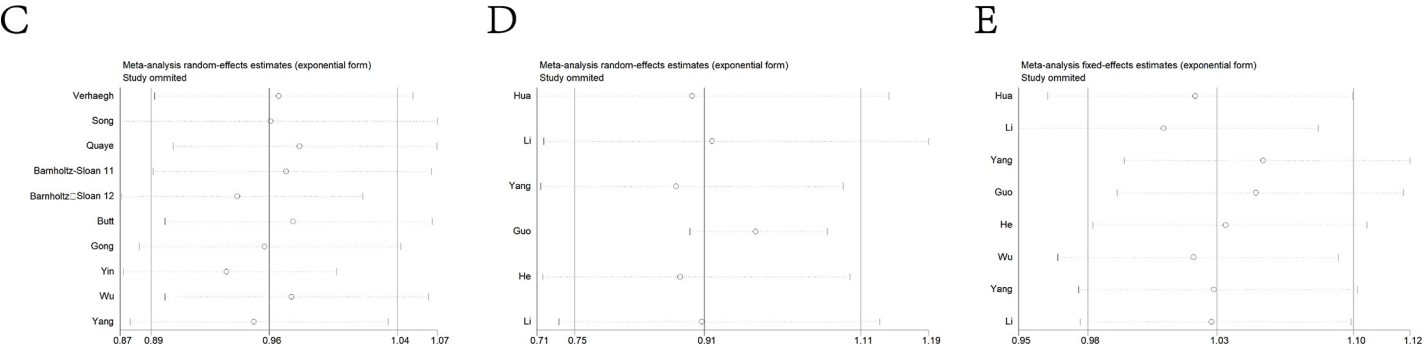

**Fig 7. Sensitivity analysis under the allele model.** A: rs2839698; B: rs217727; C: rs2107425; D: rs2735971; E: rs3024270.

associations between LncRNA H19 polymorphic variants (rs2839698, rs217727, rs2107425, rs2735971 and rs3024270) and the susceptibility to cancer were firm evidence of effect [51]. Thus lager sample size for further verification is unnecessary.

Inevitably, several additional limitations should be warranted in this meta-analysis. (1). As a multifactorial disease, overall cancers are influenced by genetic combined with environmental factors. Focusing on single gene region, this meta-analysis ignored the complex interaction between various factors, in which case the association was unilateral; (2). The amount of studies in the subgroup analysis was relatively small. Subgroups with less than three studies were retained, thus might causing the potential false associations; (3). With the limit of the study amount, subgroup analysis based on race or cancer subtypes was not performed in this article. Additionally, subgroup analysis based on sex, age and gene dosage failed to be conducted accounting for the unavailability of relevant detailed data [52]. Besides, we failed to acquire the information of details such as age and gender distribution, amount of multiple gene mutation cases and so on, in which case, multi-trait analysis seems unable to implement [53]. (4). Quality control is also one of the limitations of our study. As most of the meta-analysis, the individual study quality determines overall quality. The test of Hardy–Weinberg equilibrium was conducted in this study, results of which indicates that genotype and allele frequencies remain unchanged over the generations. Nevertheless, specific quality test could be performed [54]. (5). Causal inference analysis could be another limitation. Gene mutation can affect the occurrence and development of cancer by affecting intermediate phenotype or other exposure factors. Previous study has shown that SNPs could be of much importance in modulating some novel biomarkers. Mendelian randomization study plays a vital role in discovering the

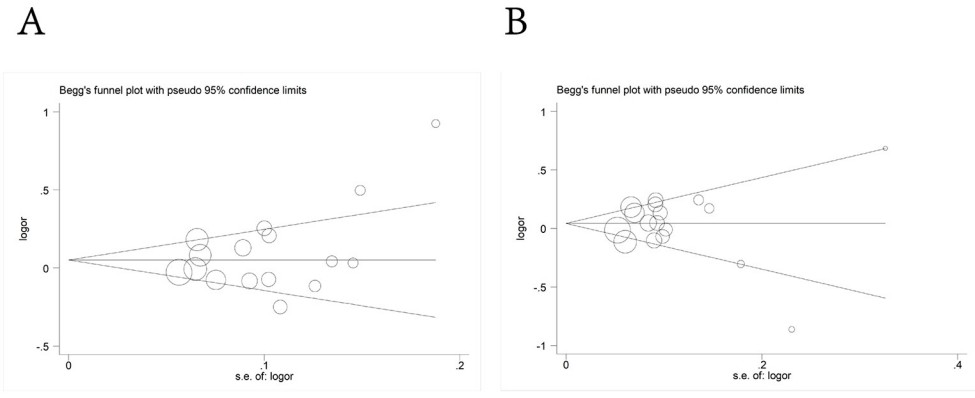

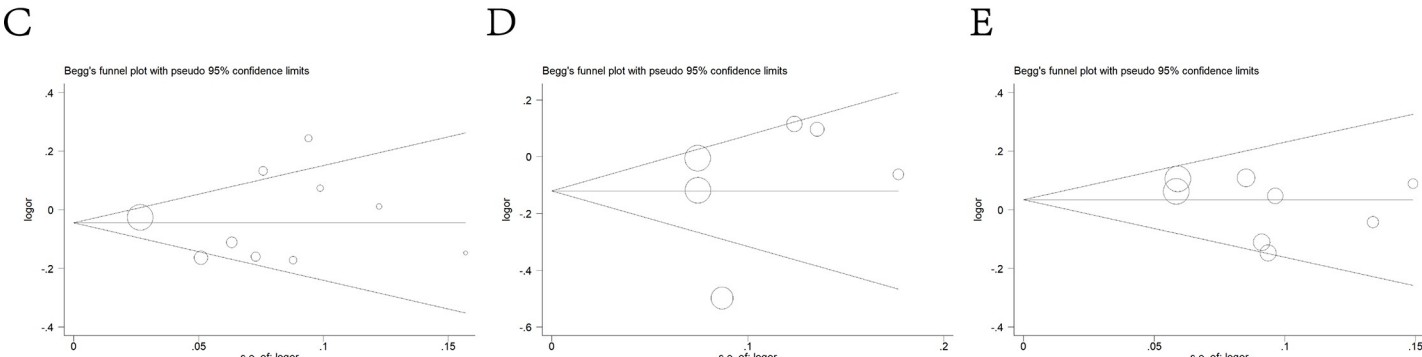

**Fig 8. Begg's funnel plot of publication bias test in the allele model.** A: rs2839698; B: rs217727; C: rs2107425; D: rs2735971; E: rs3024270.

causation of various cancers [55–57]. Conditions needs to be met for this study, while we failed to find the intermediate phenotype or other exposure factors in H19 mutation cases.

Genetic variability is significant when it comes to evaluation of disease susceptibility. In study by Allemailem at al., SNPs was helpful not only in diagnosis of prostate, but also in the further treatment for individuals [58]. Meanwhile, contributions have been made in plenty of studies retrieved in our article to testified the possibility of H19 SNPs in diagnosis and individualized treatment of various cancer. In this meta-analysis, we concentrated on the association between the overall cancer susceptibility and H19 mutation. Insufficient data of gene dosage and tumor staging from raw studies adds complications to establishing a prediction model [59, 60].

Consensus has been reached that H19 is involved in various biological process, but the potential mechanisms remain unknown. The study by Zheng at al. revealed that gene mutation can promote self adaptation [61]. On the other hand, current researches have focused on the N4-acetylcytidine on RNA, which can impact the development of cancer [62]. Thus, further exploration of according mechanism is necessary. Hence, to guaranty reliability of our meta-analysis, more large-sample, multi-center and high-quality researches should focus on the influence of different factors in the subsequent studies.

## Conclusion

To conclude, the results of this meta-analysis revealed that five H19 polymorphisms (rs2839698 G>A, rs217727 G>A, rs2107425 C>T, rs2735971 A>G and rs3024270 C>G) had

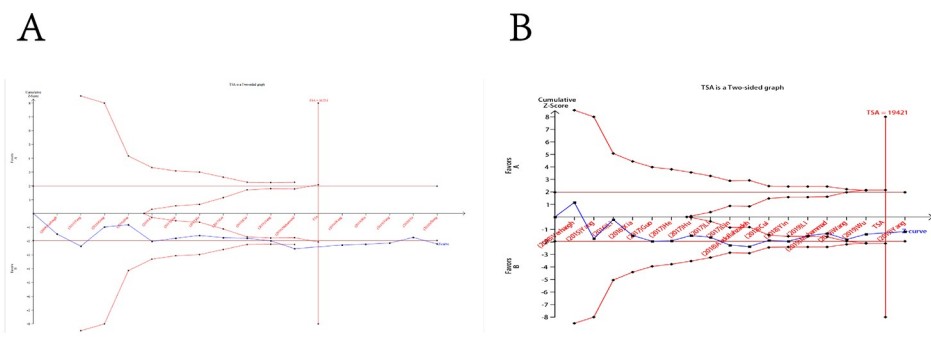

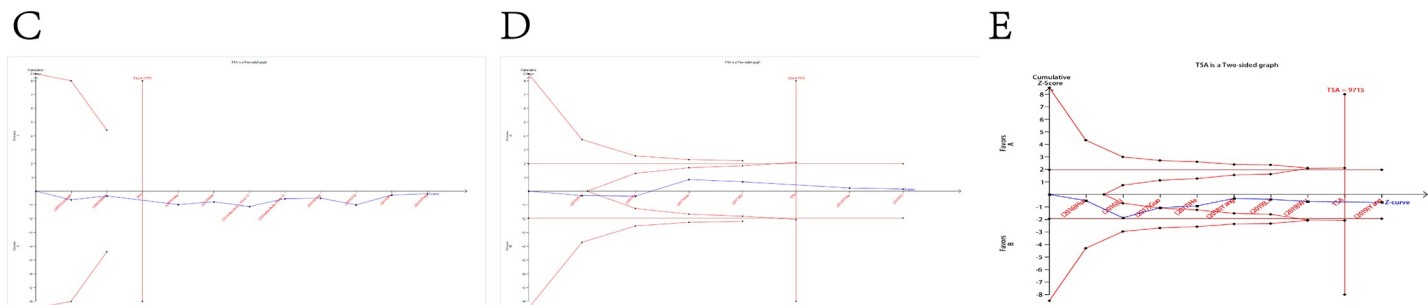

**Fig 9. Trial sequential analysis of the association between H19 polymorphisms and the susceptibility to cancer.** The required information size was calculated based on a 2-sided α = 5%, β = 15% (power 85%), and a relative risk reduction of 20%. A: rs2839698; B: rs217727; C: rs2107425; D: rs2735971; E: rs3024270.

no significant association with the overall cancer susceptibility, thereby suggesting that H19 might be not qualified for the ideal marker in the diagnosis and treatment of cancer. However, after the stratification analysis, inconsistent results still existed in different genotypic method and source of control. Thus, more high-quality studies on cancer patients of different factors were needed to confirm these findings.

## Supporting information

**S1 Checklist. PRISMA_2020_checklist.**
(DOCX)

**S1 Data. Raw data.**
(XLSX)

## Author Contributions

**Data curation:** Kunpeng Wang, Zheng Zhu, Jinbao Gu.

**Formal analysis:** Yiqiu Wang, Dayuan Zong, Peng Xue.

**Project administration:** Kunpeng Wang, Zheng Zhu, Chuanquan Tu.

**Writing – original draft:** Kunpeng Wang, Zheng Zhu, Daoyuan Lu.

**Writing – review & editing:** Kunpeng Wang, Zheng Zhu, Daoyuan Lu.

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
