## [Decision Letter · Decision Letter 0]

19 Apr 2021

PONE-D-21-10970

The influence of LncRNA H19 polymorphic variants on susceptibility to cancer: a systematic review and updated meta-analysis of 28 case-control studies

PLOS ONE

Dear Dr. Chuanquan Tu,

Thank you for submitting your manuscript to PLOS ONE. After careful consideration, we feel that it has merit but does not fully meet PLOS ONE’s publication criteria as it currently stands. Therefore, we invite you to submit a revised version of the manuscript that addresses the points raised during the review process.

We look forward to receiving your revised manuscript.

Kind regards,

Mingqing Xu

Academic Editor

PLOS ONE

Journal Requirements:

2. Please include your tables as part of your main manuscript and remove the individual files. Please note that supplementary tables should remain uploaded as separate "supporting information" files.

Reviewers' comments:

Reviewer's Responses to Questions

**Comments to the Author**

1. Is the manuscript technically sound, and do the data support the conclusions?

Reviewer #1: Yes

2. Has the statistical analysis been performed appropriately and rigorously? 

Reviewer #1: No

3. Have the authors made all data underlying the findings in their manuscript fully available?

Reviewer #1: No

4. Is the manuscript presented in an intelligible fashion and written in standard English?

Reviewer #1: No

5. Review Comments to the Author

Reviewer #1: In the manuscript entitled “The influence of LncRNA H19 polymorphic variants on susceptibility to cancer: a systematic review and updated meta-analysis of 28 case-control studies”, the authors conducted a meta-analysis to clarify the associations between LncRNA H19 polymorphic variants (rs2839698 G﹥A, rs217727 G﹥A, rs2107425 C﹥T, rs2735971 A﹥G and rs3024270 C﹥G) and the susceptibility to cancer. The novelty is limited and the scientific questions based on this meta-analysis are not addressed deeply due to lack some specific subgroup analyses. Different cancers may have different etiology, the main problem is that the authors put all cancer types together for meta-analysis.

Main cencers:

1. The following papers can be cited and followed for the meta-analytic procedures (if the data is not enough available, at least DISCUSSION should be added as the LIMITATION of this study with enough cited references to support the viewpoints):

Ref 1: Wu Y, et al. Multi-trait analysis for genome-wide association study of five psychiatric disorders. Transl Psychiatry. 2020 Jun 30;10(1):209.

Ref 2: Jiang L, et al. Sex-Specific Association of Circulating Ferritin Level and Risk of Type 2 Diabetes: A Dose-Response Meta-Analysis of Prospective Studies..J Clin Endocrinol Metab. 2019 Oct 1;104(10):4539-4551.

Ref 3: Xu M, et al. Quantitative assessment of the effect of angiotensinogen gene polymorphisms on the risk of coronary heart disease. Circulation. 2007 Sep 18;116(12):1356-66.

Trans-ethnic and trans-trait meta-analysis can be referred to Ref 1.

Subgroup analyses based on cancer subtypes, sex, age, race, gene dosage can be referred to Ref 2.

Quality control can be referred to Ref 3.

2. I am wondering if the SNPs can be used for cancer prediction, and if the authors may follow the following references to construct prediction model for cancer or cancer subtypes based on the biomarkers.

Ref 3: Yu H, et al. LEPR hypomethylation is significantly associated with gastric cancer in males.Exp Mol Pathol. 2020 Oct;116:104493. I suggest the authors cite and follow the above paper to clearly describe ROC curve for the evaluating the performences.

Ref 4: Liu M, et al. A multi-model deep convolutional neural network for automatic hippocampus segmentation and classification in Alzheimer's disease.Neuroimage. 2020 Mar;208:116459.

If machine-learning/deep learning can not be used, please discuss as the LIMITATION of this study with enough cited references to support the viewpoints.

3. I suggest that at least one paragraph in the DISCUSSION section is needed to explore the potential biological mechanisms of the identified SNPs in LncRNA H19 that can be used for the prediction of cancers, including DNA repair, cancer-related immunodeficiency, RNA modifications , LncRNA regulations, which may explain how the five SNPs in LncRNA H19 are playing roles on the overall cancer or specific cancers.

It is well know that immunology regulations may affect cancer development and people reported that Immunodeficiency may promote adaptive alterations of host gut- and tissue-based microbiome. Therefore I would suggest the authors to discuss the relationship and the potential mechanisms among Immunodeficiency, clinic-pathological information,SNP biomarkers/ LncRNA H19 gene expression, and overall survival/development of cancers. The following paper can be referred to for this discussion:

Ref 5:Zheng S, et al. Immunodeficiency Promotes Adaptive Alterations of Host Gut Microbiome: An Observational Metagenomic Study in Mice. Front Microbiol. 2019 Nov 1;10:2415. doi: 10.3389/fmicb.2019.02415. eCollection 2019.

The mRNA modifications are potentially new insights into this biological basis,especially N4-Acetylcytidine on LncRNA H19 expression. The following papers clearly disclosed the RNA abnormal expressions are mediated through RNA modifications. The recent progress in N4-Acetylcytidine on RNA expression is also playing key role on the cancer development. I suggest the authors discuss this LncRNA H19 modifications/ N4-Acetylcytidine with their findings in the DISCUSSION section. The knowledge on cancer development needs to be updated.

Ref 6: Jin G, et al. The Processing, Gene Regulation, Biological Functions, and Clinical Relevance of N4-Acetylcytidine on RNA: A Systematic Review. Mol Ther Nucleic Acids. 2020. PMID: 32171170

In the Discussion Section, the limitations and strength should be discussed in detail with enough citation, as I recommended, supporting the viewpoints. If all the above-suggested additional analyses cannot be employed for technique reasons, please discuss as the LIMITATION of this study.

Minor Concerns: The figures are not clear, and Egger’s test is better to detect publication bias. Language editing is needed.

6. PLOS authors have the option to publish the peer review history of their article (what does this mean?). If published, this will include your full peer review and any attached files.

Reviewer #1: No

---

## [Author Response · Author response to Decision Letter 0]

8 Jun 2021

Dear Editor/Reviewers：

Thank you for your prompt attention to our manuscript and helpful suggestions. We are pleased to follow your comments and the manuscript has been extensively revised according to your advice. Moreover, we have had our manuscript revised by an experienced colleague. In addition, we carefully proofread the manuscript to minimize typographical, grammatical, and bibliographical mistakes. The corresponding explanations of each point which is raised in your comments are as follows:

Reviewer #1

Question 1: The following papers can be cited and followed for the meta-analytic procedures (if the data is not enough available, at least DISCUSSION should be added as the LIMITATION of this study with enough cited references to support the viewpoints.

Ref 1: Wu Y, et al. Multi-trait analysis for genome-wide association study of five psychiatric disorders. Transl Psychiatry. 2020 Jun 30;10(1):209.

Ref 2: Jiang L, et al. Sex-Specific Association of Circulating Ferritin Level and Risk of Type 2 Diabetes: A Dose-Response Meta-Analysis of Prospective Studies..J Clin Endocrinol Metab. 2019 Oct 1;104(10):4539-4551.

Ref 3: Xu M, et al. Quantitative assessment of the effect of angiotensinogen gene polymorphisms on the risk of coronary heart disease. Circulation. 2007 Sep 18;116(12):1356-66.

Trans-ethnic and trans-trait meta-analysis can be referred to Ref 1.

Subgroup analyses based on cancer subtypes, sex, age, race, gene dosage can be referred to Ref 2.

Quality control can be referred to Ref 3.

Authors’ response: Thank you for your careful reading and helpful questions. In the previous literature study, a large number of article related to H19 polymorphisms with cancer risk were found. Therefore, we decided to use meta-analysis to put together these data in each literature for further analysis under the steps of subject determination, literature retrieval and screening, data extraction, statistical analysis and discussion. As to the study quality assessment issue, we have conducted test for HWE of each study in our meta-analysis, and results of which has been attached as supplement files in the updated Table 1. We have conducted the subgroup analysis on genotype and source of controls as shown in the article. However, with the limit of the study amount, subgroup analysis based on race or cancer subtypes was not performed in this article regretfully. Additionally, subgroup analysis based on sex, age and gene dosage failed to be conducted accounting for the unavailability of relevant detailed data. We sincerely hope that more large-sample, multi-center and high-quality researches could emerge, based on which, a systematically update could be done.

Wu Y, et al. Multi-trait analysis for genome-wide association study of five psychiatric disorders. Transl Psychiatry. 2020 Jun 30;10(1):209.

Jiang L, et al. Sex-Specific Association of Circulating Ferritin Level and Risk of Type 2 Diabetes: A Dose-Response Meta-Analysis of Prospective Studies..J Clin Endocrinol Metab. 2019 Oct 1;104(10):4539-4551.

Question 2:  I am wondering if the SNPs can be used for cancer prediction, and if the authors may follow the following references to construct prediction model for cancer or cancer subtypes based on the biomarkers.

Ref 3: Yu H, et al. LEPR hypomethylation is significantly associated with gastric cancer in males.Exp Mol Pathol. 2020 Oct;116:104493. I suggest the authors cite and follow the above paper to clearly describe ROC curve for the evaluating the performences.

Ref 4: Liu M, et al. A multi-model deep convolutional neural network for automatic hippocampus segmentation and classification in Alzheimer's disease.Neuroimage. 2020 Mar;208:116459.

If machine-learning/deep learning can not be used, please discuss as the LIMITATION of this study with enough cited references to support the viewpoints.

Authors’ response: Thanks for your considerate consideration. Indeed, describing ROC curve in order to get a specific assessment of whether the H19 SNPs could serve as a predictor of cancer is fairly attractive. Nevertheless, considering the lack of research amount upon specific cancer types, we were worried about the reliability of such data analysis. Instead of that, we referred to relevant literature and found that genetic variability is significant when it comes to evaluation of disease susceptibility. In study by Allemailem at al., SNPs was helpful not only in diagnosis of prostate, but also in the further treatment for individuals, which indicated that H19 SNPs also have the potential ability of predicting the cancer development. Meanwhile, contributions have been made in plenty of studies retrieved in our article to testified the possibility of H19 SNPs in diagnosis and individualized treatment of various cancer.

53 Allemailem, K. S. et al., Single nucleotide polymorphisms (SNPs) in prostate cancer: its implications in diagnostics and therapeutics. AM J TRANSL RES 13 3868 (2021).

Question 3: I suggest that at least one paragraph in the DISCUSSION section is needed to explore the potential biological mechanisms of the identified SNPs in LncRNA H19 that can be used for the prediction of cancers, including DNA repair, cancer-related immunodeficiency, RNA modifications , LncRNA regulations, which may explain how the five SNPs in LncRNA H19 are playing roles on the overall cancer or specific cancers.

It is well know that immunology regulations may affect cancer development and people reported that Immunodeficiency may promote adaptive alterations of host gut- and tissue-based microbiome. Therefore I would suggest the authors to discuss the relationship and the potential mechanisms among Immunodeficiency, clinic-pathological information,SNP biomarkers/ LncRNA H19 gene expression, and overall survival/development of cancers. The following paper can be referred to for this discussion:

Ref 5:Zheng S, et al. Immunodeficiency Promotes Adaptive Alterations of Host Gut Microbiome: An Observational Metagenomic Study in Mice. Front Microbiol. 2019 Nov 1;10:2415. doi: 10.3389/fmicb.2019.02415. eCollection 2019.

The mRNA modifications are potentially new insights into this biological basis,especially N4-Acetylcytidine on LncRNA H19 expression. The following papers clearly disclosed the RNA abnormal expressions are mediated through RNA modifications. The recent progress in N4-Acetylcytidine on RNA expression is also playing key role on the cancer development. I suggest the authors discuss this LncRNA H19 modifications/ N4-Acetylcytidine with their findings in the DISCUSSION section. The knowledge on cancer development needs to be updated.

Ref 6: Jin G, et al. The Processing, Gene Regulation, Biological Functions, and Clinical Relevance of N4-Acetylcytidine on RNA: A Systematic Review. Mol Ther Nucleic Acids. 2020. PMID: 32171170

In the Discussion Section, the limitations and strength should be discussed in detail with enough citation, as I recommended, supporting the viewpoints. If all the above-suggested additional analyses cannot be employed for technique reasons, please discuss as the LIMITATION of this study.

Authors’ response: Thank you for your careful and wise suggestions. According to your advice, we have carefully read your opinions and had a heated discussion by all co-authors so as to reach a unified point of view. After the discussion, we searched studies on the mechanisms of the the identified SNPs in LncRNA H19. With detailed reading, we acquired plenty of precious information and hint upon the potential mechanisms of how H19 SNPs indeed influence cancer development. The study by Zheng at al. performed a research of immunodeficiency mice, and assessment of the microbiome of their fecal samples revealed that gene mutation can promote self adaptation. As you mentioned in the advice, immunology regulations may affect cancer development by promoting adaptive alterations of host gut- and tissue-based microbiome. We would like to pay more attention to this direction as soon as we obtain relevant pathological information in detail. Additionally, current researches have also focused on the N4-acetylcytidine on RNA, which can impact the development of cancer. However, related experiments and researches have not been conducted on LncRNA frequently, especially on H19 SNPs. Thus, we felt the urge of filling the blank of this field and further exploration of according mechanism is necessary. Besides, some the figures are too big to uploaded, so we just uploaded their screenshots. But we could surely provide the original version if necessary. 

54 Zheng, S. et al., Immunodeficiency Promotes Adaptive Alterations of Host Gut Microbiome: An Observational Metagenomic Study in Mice. FRONT MICROBIOL 10 2415 (2019).

55 Jin, G., Xu, M., Zou, M. & Duan, S., The Processing, Gene Regulation, Biological Functions, and Clinical Relevance of N4-Acetylcytidine on RNA: A Systematic Review. Mol Ther Nucleic Acids 20 13 (2020).

Yours sincerely

Chuanquan Tu

Department of Urology, The First People’s Hospital of Lianyungang, Lianyungang Clinical Medical College of Nanjing Medical University, Lianyungang, 222002, China.

2021-05-30

---

## [Decision Letter · Decision Letter 1]

18 Jun 2021

PONE-D-21-10970R1

The influence of LncRNA H19 polymorphic variants on susceptibility to cancer: a systematic review and updated meta-analysis of 28 case-control studies

PLOS ONE

Dear Dr. Tu,

Thank you for submitting your manuscript to PLOS ONE. After careful consideration, we feel that it has merit but does not fully meet PLOS ONE’s publication criteria as it currently stands. Therefore, we invite you to submit a revised version of the manuscript that addresses the points raised during the review process.

The quality is improved so much according to the comments. Some minor concerns can be considered for further revision before publishing this paper.  Please revise carefully because we usually only allows up to three-round revisions.

We look forward to receiving your revised manuscript.

Kind regards,

Mingqing Xu

Academic Editor

PLOS ONE

Journal Requirements:

Additional Editor Comments (if provided):

The quality is improved so much according to the comments. Some minor concerns can be considered for further revision before publishing this paper.

Reviewers' comments:

Reviewer's Responses to Questions

**Comments to the Author**

1. If the authors have adequately addressed your comments raised in a previous round of review and you feel that this manuscript is now acceptable for publication, you may indicate that here to bypass the “Comments to the Author” section, enter your conflict of interest statement in the “Confidential to Editor” section, and submit your "Accept" recommendation.

Reviewer #1: All comments have been addressed

Reviewer #2: (No Response)

2. Is the manuscript technically sound, and do the data support the conclusions?

Reviewer #1: No

Reviewer #2: No

3. Has the statistical analysis been performed appropriately and rigorously? 

Reviewer #1: No

Reviewer #2: No

4. Have the authors made all data underlying the findings in their manuscript fully available?

Reviewer #1: No

Reviewer #2: Yes

5. Is the manuscript presented in an intelligible fashion and written in standard English?

Reviewer #1: Yes

Reviewer #2: Yes

6. Review Comments to the Author

Reviewer #1: For the paper entitled “The influence of LncRNA H19 polymorphic variants on susceptibility to cancer: a systematic review and updated meta-analysis of 28 case-control studies”

My previous concerns are not well addressed or followed for additional analyses. My additional comments are as follows:

1. The following papers can be cited and followed for the meta-analytic procedures (if the data is not enough available, at least DISCUSSION should be added as the LIMITATION of this study with enough cited references to support the viewpoints):

Reference 1: Wu Y, et al. Multi-trait analysis for genome-wide association study of five psychiatric disorders. Transl Psychiatry. 2020 Jun 30;10(1):209.

Reference 2: Jiang L, et al. Sex-Specific Association of Circulating Ferritin Level and Risk of Type 2 Diabetes: A Dose-Response Meta-Analysis of Prospective Studies..J Clin Endocrinol Metab. 2019 Oct 1;104(10):4539-4551.

Reference 3: Xu M, et al. Quantitative assessment of the effect of angiotensinogen gene polymorphisms on the risk of coronary heart disease. Circulation. 2007 Sep 18;116(12):1356-66.

Trans-ethnic and trans-trait meta-analysis can be referred to Ref 1.

Subgroup analyses based on cancer subtypes, sex, age, race, gene dosage can be referred to Ref 2.

Quality control can be referred to Ref 3.

2. I am wondering if the SNPs can be used for cancer prediction, and if the authors may follow the following references to construct prediction model for cancer or cancer subtypes based on the biomarkers.The authors may discuss the possibility to use the genetic variants related to cancer or cancer subtypes for the prediction or early diagnosis of Cancer. The authors may cite the following papers for discussion or follow the analytic procedures to construct machine learning prediction models.

Reference 1:Yu H, et al. LEPR hypomethylation is significantly associated with gastric cancer in males.Exp Mol Pathol. 2020 Oct;116:104493.

Reference 2:Liu M, et al. A multi-model deep convolutional neural network for automatic hippocampus segmentation and classification in Alzheimer's disease.Neuroimage. 2020 Mar;208:11645

2. I am wondering if the authors may Integrate genotype data with eQTL from GTEX or pQTLs is to explore the causality of the LncRNA H19 genetic variants in the development of Cancer. But I strong suggest to do causal inference analysis to see if the LncRNA H19 genetic variants in this gene are causally triggering the development of Cancer through mediating the expression of this gene in specific cancer tissues. If cannot, please discuss the limitations in the Discussion in detail with additional citations to support the viewpoints. For these reasons, the following papers regarding causal inference between genetic varients,inter-mediator phenotype,like LncRNA expression, and disease outcome can be cited and followed.

Reference 1: Fuquan Zhang, Ancha Baranova, Chao Zhou, et al. Causal influences of neuroticism on mental health and cardiovascular disease. Human Genetics. 2021 May 1

Reference 2: Fuquan Zhang, et al. Genetic evidence suggests posttraumatic stress disorder as a subtype of major depressive disorder. Journal of Clinical Investigation. 2021 May 30

Reference 3: Xinhui Wang, et al. Genetic support of a causal relationship between iron status and type 2 diabetes: a Mendelian randomization study. The Journal of Clinical Endocrinology & Metabolism. In Press.

Reviewer #2: This paper explored the five H19 326 polymorphisms (rs2839698 G﹥A, rs217727 G﹥A, rs2107425 C﹥T, rs2735971 A﹥ 327 G and rs3024270 C﹥G) with overall cancer 328 susceptibility by use of meta-analysis. Different cancers have different etiology and genetic basis. I suggest the author to try to do stratified the meta-analysis by different cancer subtypes, like lung cancer, breast cancer,etc. or at least discuss this limitation.

The figures are not clear and readable.

7. PLOS authors have the option to publish the peer review history of their article (what does this mean?). If published, this will include your full peer review and any attached files.

Reviewer #1: No

Reviewer #2: No

---

## [Author Response · Author response to Decision Letter 1]

4 Jul 2021

Dear Editor/Reviewers：

Thank you for your prompt attention to our manuscript and helpful suggestions. We are pleased to follow your comments and the manuscript has been extensively revised according to your advice. Moreover, we have had our manuscript revised by an experienced colleague. In addition, we carefully proofread the manuscript to minimize typographical, grammatical, and bibliographical mistakes. The corresponding explanations of each point which is raised in your comments are as follows:

Reviewer #1

Question 1: The following papers can be cited and followed for the meta-analytic procedures (if the data is not enough available, at least DISCUSSION should be added as the LIMITATION of this study with enough cited references to support the viewpoints):

Reference 1: Wu Y, et al. Multi-trait analysis for genome-wide association study of five psychiatric disorders. Transl Psychiatry. 2020 Jun 30;10(1):209.

Reference 2: Jiang L, et al. Sex-Specific Association of Circulating Ferritin Level and Risk of Type 2 Diabetes: A Dose-Response Meta-Analysis of Prospective Studies..J Clin Endocrinol Metab. 2019 Oct 1;104(10):4539-4551.

Reference 3: Xu M, et al. Quantitative assessment of the effect of angiotensinogen gene polymorphisms on the risk of coronary heart disease. Circulation. 2007 Sep 18;116(12):1356-66.

Authors’ response: Thank you for your careful reading and helpful advices again. In the previous literature study, a large number of article related to H19 polymorphisms with cancer risk were found. Therefore, we decided to use meta-analysis to put together these data in each literature for further analysis under the steps of subject determination, literature retrieval and screening, data extraction, statistical analysis and discussion. During the data extraction, we failed to acquire the information of details such as age and gender distribution, amount of multiple gene mutation cases and so on, in which case, multi-trait analysis seems unable to implement. However, we do consider it as a potential research direction for further investigation. As for subgroup analysis, in fact, with the limit of the study amount, subgroup analysis based on race (Caucasion:7; African:1; Asian:49) or cancer subtypes (Bladder cancer:13; Breast cancer:11; Lung cancer:7; Hepatocellular cancer:7; Colorectal cancer:5; Ovarian cancer:3; Oral squamous cell carcinoma:4; Osteosarcoma:4; Gastric cancer:2: Pancreatic cancer:1) was not performed in this article regretfully. Instead, we have conducted the subgroup analysis on genotype and source of controls as shown in the article. Additionally, detailed data of sex, age and gene dosage were the unavailable either. We sincerely hope that more large-sample, multi-center and high-quality researches could emerge, based on which, a systematically update could be done. As to the study quality assessment issue, we have conducted test for HWE of each study in our meta-analysis, and results of which has been attached as supplement files in the updated Table 1. 

Wu Y, et al. Multi-trait analysis for genome-wide association study of five psychiatric disorders. Transl Psychiatry. 2020 Jun 30;10(1):209.

Jiang L, et al. Sex-Specific Association of Circulating Ferritin Level and Risk of Type 2 Diabetes: A Dose-Response Meta-Analysis of Prospective Studies..J Clin Endocrinol Metab. 2019 Oct 1;104(10):4539-4551.

Xu M, et al. Quantitative assessment of the effect of angiotensinogen gene polymorphisms on the risk of coronary heart disease. Circulation. 2007 Sep 18;116(12):1356-66.

Question 2:   I am wondering if the SNPs can be used for cancer prediction, and if the authors may follow the following references to construct prediction model for cancer or cancer subtypes based on the biomarkers.The authors may discuss the possibility to use the genetic variants related to cancer or cancer subtypes for the prediction or early diagnosis of Cancer. The authors may cite the following papers for discussion or follow the analytic procedures to construct machine learning prediction models.

Reference 1:Yu H, et al. LEPR hypomethylation is significantly associated with gastric cancer in males.Exp Mol Pathol. 2020 Oct;116:104493.

Reference 2:Liu M, et al. A multi-model deep convolutional neural network for automatic hippocampus segmentation and classification in Alzheimer's disease.Neuroimage. 2020 Mar;208:11645

Authors’ response: Thanks! We appreciated for your considerate consideration a lot. In study by Allemailem at al., SNPs was helpful not only in diagnosis of prostate, but also in the further treatment for individuals, which indicated that H19 SNPs also have the potential ability of predicting the cancer development. Meanwhile, contributions have been made in plenty of studies retrieved in our article to testified the possibility of H19 SNPs in diagnosis and individualized treatment of various cancer. In this meta analysis, we concentrated on the association between the overall cancer susceptibility and H19 mutation. Insufficient data of gene dosage and tumor staging from raw studies adds complications to establishing a prediction model. But as compensation, sensitive analysis was used to examine the stability and reliability of the results through recalculating the pooled ORs following the sequential exclusion of a single study at a time.

Indeed, describing ROC curve in order to get a specific assessment of whether the H19 SNPs could serve as a predictor of cancer is fairly attractive. Nevertheless, considering the lack of research amount upon specific cancer types, we were worried about the reliability of such data analysis. In the mean time, although pathological biopsy could be the gold standard for the diagnosis of most tumors, other diagnostic examination vary a lot due to different cancer types. In that case, lack of comparison makes the results ROC curve showing less meaningful. By that consideration, we referred to relevant literature and found that genetic variability is significant when it comes to evaluation of disease susceptibility. 

Allemailem, K. S. et al., Single nucleotide polymorphisms (SNPs) in prostate cancer: its implications in diagnostics and therapeutics. AM J TRANSL RES 13 3868 (2021).

Yu H, et al. LEPR hypomethylation is significantly associated with gastric cancer in males.Exp Mol Pathol. 2020 Oct;116:104493.

Liu M, et al. A multi-model deep convolutional neural network for automatic hippocampus segmentation and classification in Alzheimer's disease.Neuroimage. 2020 Mar;208:11645

Question 3: I am wondering if the authors may Integrate genotype data with eQTL from GTEX or pQTLs is to explore the causality of the LncRNA H19 genetic variants in the development of Cancer. But I strong suggest to do causal inference analysis to see if the LncRNA H19 genetic variants in this gene are causally triggering the development of Cancer through mediating the expression of this gene in specific cancer tissues. If cannot, please discuss the limitations in the Discussion in detail with additional citations to support the viewpoints. For these reasons, the following papers regarding causal inference between genetic varients,inter-mediator phenotype,like LncRNA expression, and disease outcome can be cited and followed.

Reference 1: Fuquan Zhang, Ancha Baranova, Chao Zhou, et al. Causal influences of neuroticism on mental health and cardiovascular disease. Human Genetics. 2021 May 1

Reference 2: Fuquan Zhang, et al. Genetic evidence suggests posttraumatic stress disorder as a subtype of major depressive disorder. Journal of Clinical Investigation. 2021 May 30

Reference 3: Xinhui Wang, et al. Genetic support of a causal relationship between iron status and type 2 diabetes: a Mendelian randomization study. The Journal of Clinical Endocrinology & Metabolism. In Press.

Authors’ response: Thank you for your careful and wise suggestions. According to your advice, we have carefully read your opinions and had a heated discussion by all co-authors so as to reach a unified point of view. As we know, gene mutation can affect the occurrence and development of cancer by affecting intermediate phenotype or other exposure factors. Previous study has shown that SNPs could be of much importance in modulating some novel biomarkers. Mendelian randomization study plays a vital role in discovering the causation of various cancers. Conditions needs to be met for this study, while we failed to find the intermediate phenotype or other exposure factors in H19 mutation cases. But still, genetic causal analysis inspires us for future experiments.

We also searched studies on the causation studies of the the identified SNPs in LncRNA H19. With detailed reading, we acquired plenty of precious information and hint upon the potential the issue mentioned above. Although the relevant eQTL information is easily available from database, gene dosage and tumor staging of cases we work on are all unavailable in raw studies. So we estimated the strength of the associations between the H19 polymorphisms and cancer susceptibility, using the pooled odds ratios (ORs) with 95% confidence intervals (CIs), applying five main genetic comparison models: allele model, homozygous model, heterozygous model, dominant model and recessive model.

Allemailem, K. S. et al., Single nucleotide polymorphisms (SNPs) in prostate cancer: its implications in diagnostics and therapeutics. AM J TRANSL RES 13 3868 (2021).

Fuquan Zhang, Ancha Baranova, Chao Zhou, et al. Causal influences of neuroticism on mental health and cardiovascular disease. Human Genetics. 2021 May 1

Fuquan Zhang, et al. Genetic evidence suggests posttraumatic stress disorder as a subtype of major depressive disorder. Journal of Clinical Investigation. 2021 May 30

Xinhui Wang, et al. Genetic support of a causal relationship between iron status and type 2 diabetes: a Mendelian randomization study. The Journal of Clinical Endocrinology & Metabolism. In Press.

Reviewer #2

Question 1: This paper explored the five H19 326 polymorphisms (rs2839698 G﹥A, rs217727 G﹥A, rs2107425 C﹥T, rs2735971 A﹥ 327 G and rs3024270 C﹥G) with overall cancer 328 susceptibility by use of meta-analysis. Different cancers have different etiology and genetic basis. I suggest the author to try to do stratified the meta-analysis by different cancer subtypes, like lung cancer, breast cancer,etc. or at least discuss this limitation.

Authors’ response: Thank you for your careful and wise suggestions. Although myriad researches upon the associations between LncRNA H19 polymorphic variants (rs2839698 G﹥A, rs217727 G﹥A, rs2107425 C﹥T, rs2735971 A﹥G and rs3024270 C﹥G) and the susceptibility to cancer have been conducted, results remained contradictory and perplexing. So this meta-analysis aimed at deriving a more accurate evaluation in all relevant published studies of the such associations. For further research, we have conducted the subgroup analysis on genotype and source of controls as shown in the article. When it comes to more subgroup analysis, detailed data of sex, age and gene dosage were the unavailable. Also, with the limit of the study amount, subgroup analysis based on race(Caucasion:7; African:1; Asian:49) or cancer subtypes(Bladder cancer:13; Breast cancer:11; Lung cancer:7; Hepatocellular cancer:7; Colorectal cancer:5; Ovarian cancer:3; Oral squamous cell carcinoma:4; Osteosarcoma:4; Gastric cancer:2: Pancreatic cancer:1) was not performed in this article regretfully. We sincerely hope that more large-sample, multi-center and high-quality researches could emerge, based on which, a systematically update could be done.

Yours sincerely

Chuanquan Tu

Department of Urology, The First People’s Hospital of Lianyungang, Lianyungang Clinical Medical College of Nanjing Medical University, Lianyungang, 222002, China.

2021-06-24

---

## [Editor Report · Decision Letter 2]

7 Jul 2021

The influence of LncRNA H19 polymorphic variants on susceptibility to cancer: a systematic review and updated meta-analysis of 28 case-control studies

PONE-D-21-10970R2

Dear Dr. TU,

We’re pleased to inform you that your manuscript has been judged scientifically suitable for publication and will be formally accepted for publication once it meets all outstanding technical requirements.

Kind regards,

Mingqing Xu

Academic Editor

PLOS ONE

Additional Editor Comments (optional):

It can be accepted for publishing now.
---

## [Editor Report · Acceptance letter]

15 Jul 2021

PONE-D-21-10970R2 

The influence of LncRNA H19 polymorphic variants on susceptibility to cancer : a systematic review and updated meta-analysis of 28 case-control studies 

Dear Dr. Tu:

I'm pleased to inform you that your manuscript has been deemed suitable for publication in PLOS ONE. Congratulations! Your manuscript is now with our production department. 

Kind regards, 

on behalf of

Dr. Mingqing Xu 

Academic Editor

PLOS ONE